

# Improving the inter-hemispheric gradient of total column atmospheric $CO_2$ and $CH_4$ in simulations with the ECMWF semi-Lagrangian atmospheric global model

Anna Agusti-Panareda[1], Michail Diamantakis[1], Victor Bayona[1], Friedrich Klappenbach[2], and Andre Butz[2]

[1]European Centre for Medium-Range Weather Forecasts, Reading, United Kingdom
[2]IMK-ASF, Karlsruhe Institute of Technology (KIT), Leopoldshafen, Germany

*Correspondence to:* Anna Agusti-Panareda
(Anna.Agusti-Panareda@ecmwf.int)

**Abstract.** It is a widely established fact that standard semi-Lagrangian advection schemes are highly efficient numerical techniques for simulating the transport of atmospheric tracers. However, as they are not formally mass conserving, it is essential to use some method for restoring mass conservation in long time range forecasts. A common approach is to use global mass fixers. This is the case of the semi-Lagrangian advection scheme in the Integrated Forecasting System (IFS) model used by the

Copernicus Atmosphere Monitoring Service (CAMS) at the European Centre for Medium range Weather Forecasts (ECMWF).

Mass fixers are algorithms with substantial differences in complexity and sophistication but in general of low computational cost. This paper shows the positive impact mass fixers have on the inter-hemispheric gradient of total atmospheric column averaged $CO_2$ and $CH_4$, a crucial feature of their spatial distribution. Two algorithms are compared: the simple "proportional" and the more complex Bermejo & Conde schemes. The former is widely used by several Earth system climate models as well

the CAMS global forecasts and analysis of atmospheric composition while the latter has been recently implemented in IFS. Comparisons against total column observations demonstrate that the proportional mass fixer is shown to be suitable for the low resolution simulations but for the high resolution simulations the Bermejo & Conde scheme gives clearly better results. These results have potential repercussions for climate Earth system models using proportional mass fixers as their resolution increases. It also emphasizes the importance of benchmarking the tracer mass fixers with the inter-hemispheric gradient of

long-lived greenhouse gases using observations.

## 1  Introduction

The monitoring and prediction of climate change relies on modelling accurately the long-lived greenhouse gases using Earth system models (ESM) (e.g. Jones et al., 2013; Keppel-Aleks et al., 2013). Carbon dioxide ($CO_2$) and methane ($CH_4$) are the most important anthropogenic greenhouse gases (Forster et al., 2007). Because of their relevance to climate mitigation and

policy making, they are monitored using flux inversion systems based on atmospheric chemical transport models (CTM) (e.g. Gurney et al., 2002; Kirschke et al., 2013). Complementing the climate monitoring, global analyses and forecasts of $CO_2$ and





$CH_4$ are also performed each day as part of the Copernicus Atmosphere Monitoring Service (CAMS) (Agusti-Panareda et al., 2014; Massart et al., 2014) at the European Centre for Medium-range Weather Forecasts (ECMWF) using the Integrated Forecasting System (IFS, www.ecmwf.int/en/forecasts/documentation-and-support/changes-ecmwf-model/ifs-documentation).

Both atmospheric $CO_2$ and $CH_4$ are characterised by a trend associated with an annual growth rate, a seasonal cycle and an
inter-hemispheric gradient, which is consistent with the temporal and spatial distribution of their sources and sinks, tropopause height and atmospheric transport (Keppel-Aleks et al., 2011; Saito et al., 2012). In ESMs and CTMs the transport is modelled using advection, convection and turbulent mixing schemes based on Numerical Weather Prediction (NWP) methods. The semi-Lagrangian (SL) advection scheme is widely used in NWP (e.g. the ECMWF IFS model, Environment Canada GEM model, de Grandpré et al. (2016)) and ESMs (e.g. ACCESS, HadGM2, documented by Corbin and Law, 2011; Collins et al., 2011)
because of its high numerical stability, accuracy and computational efficiency. Furthermore, for the problem of multiple tracer advection, it is undeniably the most efficient approach given that for each tracer the transport operation reduces to interpolating the field from the fixed grid to the time-step dependent departure point grid. The latter is re-computed only once at each new time-step which implies that the same interpolation weights can be used for all tracers (and prognostic fields in general). However, the non-flux form of the SL scheme by default does not conserve mass. This can lead to small errors in the global
mass of tracers when modelling the tracer advection. In the case of $CH_4$ and $CO_2$, these errors accumulate with time because there is a slow or non-existent chemical sink in the atmosphere. It is therefore imperative to apply a mass fixer in order to restore the conservation of the total tracer mass. This is particularly important for $CO_2$, as the mass conservation error can reach values that are as large as the observed global mass trend resulting from their surface fluxes and can significantly distort its large-scale distribution (e.g. Houweling et al., 2010). There are several methods to fix the global tracer mass, from the simple proportional
mass fixers to more sophisticated algorithms that focus the correction where the conservation error associated with the tracer advection is assumed to be largest, i.e. in the regions with strongest gradients. Because of its simplicity, the proportional mass fixer is widely used by ESMs and NWP models (Collins et al., 2011; Corbin and Law, 2011; Agusti-Panareda et al., 2014; Flemming et al., 2015). There are different implementations of the global proportional mass fixer. However, the correction procedure is very homogeneous/uniform. For this reason, it is prone to the artificial transfer of mass and long-range propagation
of errors. Therefore, it has the potential to create a distortion in the inter-hemispheric gradient of tracers (Maksyutov et al., 2008).

Diamantakis and Flemming (2014) implemented and tested several of these global mass fixers on the humidity, cloud fields and ozone in the IFS. Both $CO_2$ and $CH_4$ have different characteristics and requirements than shorter-lived reactive gases and humidity. Because of their long life, they are generally well-mixed with smooth gradients, and large background values
relative to their gradients. Their large-scale spatial variability is characterized by a relatively weak inter-hemispheric gradient (of the order of 100 ppb or 5% for $CH_4$ and 10 ppm or 2.5% for $CO_2$). Nevertheless, it constitutes a crucial feature to represent in the models because it reflects the spatial distribution of the surface sources/sinks (Dargaville et al., 2003; Patra et al., 2011). Considering these properties and the computational cost, flexibility and efficiency, the Bermejo and Conde (2002) fixer is deemed to be the most suited among the available schemes in the IFS for the modelling requirements of the long-





lived greenhouse gases. This is consistent with the recent tests performed with the Canadian Semi-Langrangian model by de Grandpré et al. (2016) and Polavarapu et al. (2016).

This paper presents a comparison of a taylored Bermejo and Conde (2002) mass fixer and the proportional mass fixer, which was operational until recently in the CAMS $CO_2$ and $CH_4$ forecasting and analysis system (Agusti-Panareda et al., 2014;

Massart et al., 2014) and it is also widely used in Earth system climate models (Corbin and Law, 2011; Collins et al., 2011; Jones et al., 2011). The impact of the two mass fixers on the preservation of the $CO_2$ and $CH_4$ inter-hemispheric gradient is a crucial benchmark for testing their suitability in any $CO_2$ and $CH_4$ forecasting and analysis system. Furthermore, this study can provide valuable feedback to the Earth system climate models using the simple global proportional mass fixer. The impact of resolution on the mass conservation and performace of the mass fixers can help guide the choice of mass fixer in future

climate simulations.

The structure the the paper is as follows: in Sect. 2 the mass fixers are described; the experiments performed to test the impact of the mass fixers are presented in Sect. 3; the observations are documented in Sect. 4; the results from the experiments and their evaluation using observations are provided in Sect. 5; a summary of the main findings is given in Sect. 6.

## 2   Global tracer mass fixers

The two tracer mass fixers selected in this study are described in this section. The algorithms of these fixers are described in detail by Diamantakis and Flemming (2014) as part of a set available in the ECMWF IFS model. Thus, their notation is used henceforth. A few minor modifications have been necessary in order to fine tune these algorithms for simulating the transport of long-lived greenhouse gases. For example, it was found that, given that a mass mixing ratio formulation is used, a small mass conservation error in the total atmospheric mass after advection can lead to a systematic accumulation of the tracer mass

conservation error with time. This stems from the fact that the global mass of a tracer is computed using surface pressure (see Eq. (1) below), the mass conservation error always has the same sign and finally, there are no atmospheric processes (e.g. strong chemical sources/sinks) that can counter the effect of the systematic error accumulation. It was therefore necessary to apply the mass fixer on surface pressure as well, as explained in the paragraphs below.

The IFS is a hydrostatic model using a pressure based coordinate system which implies that the surface pressure field is

required to compute the total tracer mass. For example, the mass of a tracer $\chi$ with mass mixing ratio $\phi_\chi = \rho_\chi/\rho$, where $\rho_\chi$, $\rho$ the tracer and air-density respectively, is given by:

$$M(\phi_\chi, p) = \sum_{j=1}^{N} A_j \sum_{k=1}^{K} \phi_{jk} \frac{\Delta p_{jk}}{g} \qquad (1)$$

where $p$ is the atmospheric pressure field, $A_j$ is the horizontal surface area of box $j$, $k$ is the vertical model level and $g$ the gravitational constant. Each model level consists of $N$ grid-points and there are $K$ vertical levels.





Experiments with IFS at different resolutions showed that it is important that after the advection step and before the mass of the tracer is corrected, the pressure field needs to be corrected in order to ensure that the total mass of air

$$M(p) = \sum_{j=1}^{N} A_j \sum_{k=1}^{K} \frac{\Delta p_{jk}}{g} \qquad (2)$$

is globally conserved in the tracer mass computation. We did not find large differences in the method of correction applied here and this can be done either by the proportional algorithm (described below) or by the McGregor scheme described also in Diamantakis and Flemming (2014). The latter was chosen as it gives realistic corrections of surface pressure in regions with cyclonic activity or regions with orography and additionally has very low computational cost. For a model using a height based vertical coordinate system and density as the prognostic variable, the correction should be applied on density. In the following sections, the pressure after the SL advection is always corrected to have the same global value as before advection by using the proportional fixer presented below.

## 2.1 Global proportional mass fixer

The proportional mass fixer only requires the computation of the total tracer mass before and after the SL advection step. The mixing ratio of every single grid-point is then multiplied by the same scaling factor i.e.

$$(\phi_\chi)_{jk} = \alpha \left( \phi_\chi^* \right)_{jk}, \qquad \alpha = \frac{M(\phi_\chi^0, p^0)}{M(\phi_\chi^*, p^*)}$$

where $(\phi_\chi^0, p^0)$, $(\phi_\chi^*, p^*)$ the tracer mixing ratio and the pressure field before and after the SL advection step respectively. Long-lived tracers also require the correction of the pressure field to ensure global mass conservation of air before computing the scaling factor $\alpha$, as already discussed at the beginning of section 2. The advantages of this fixer is that it is computationally cheap, it is easy to implement, it preserves positive definiteness and for tracers such as $CO_2$ and $CH_4$, it produces very small increments. The disadvantage is that the mass of every grid-point is adjusted by the same factor implying that regions with large transport and mass conservation error are corrected by an equal proportion with regions where these errors are small and therefore the solution deteriorates there. This scheme is used by the ACCESS (Corbin and Law, 2011) and HadGEM-2 (Collins et al., 2011; Jones et al., 2011) Earth System Climate models.

## 2.2 Bermejo & Conde mass fixer

A 3D version of the Bermejo and Conde (2002) mass fixer has been implemented in the IFS (Diamantakis and Flemming, 2014) that provides an effective alternative to the proportional global mass fixer for the simulation of long-lived greenhouse gases. This scheme preserves the monotonicity of an advected field (provided the original field is also monotone) and overall the increments it computes are small. A weighted approach is used where a different weighting factor is applied when correcting the mass mixing ratios of different grid-points. For grid-points in regions where the field is smooth the weights are very small and the correction is negligible while for grid-points in regions with large gradients the weight and therefore the computed increments are larger. This is the major advantage of this method which is well suited for simulating the transport of long lived





gases such as $CO_2$ and $CH_4$. These species are spread everywhere in the globe being fairly uniform in some geographical regions (e.g. Antarctica) while they have considerable gradients in other regions (e.g. Africa, South America). Furthermore, the mass conserving field the scheme computes has minimum distance from the original advected non-conserving field as it is the solution of a minimization problem which ensures that the increments are overall small.

Using the notation of the previous section and ignoring for simplicity the subscript $\chi$, the correction the Bermejo & Conde scheme introduces to the grid-point mixing ratio in IFS can be written as

$$\phi_{jk} = \phi_{jk}^* - \lambda w_{jk}, \quad \lambda = \frac{\delta M}{\displaystyle\sum_{j=1}^{N} A_j \sum_{k=1}^{K} w_{jk} \frac{\Delta p_{jk}^*}{g}}, \tag{3}$$

where

$$\delta M = M(\phi_\chi^*, p^*) - M(\phi_\chi^0, p^0)$$

is the small global mass error. In this case, we have chosen

$$w_{jk} = \max\left[0, sgn(\delta M) sgn\left(\phi_{jk}^* - \phi_{jk}^L\right) \left|\phi_{jk}^* - \phi_{jk}^L\right|^\beta \frac{p_{jk}}{p_{j0}}\right], \tag{4}$$

which depends on the difference between the cubic interpolated field $\phi^*$ and the linear one $\phi^L$ as described in Diamantakis and Flemming (2014). It was argued there that an appropriate setting for the parameter $\beta$ would be 1. This conclusion was based on testing done with moist and fast chemically active tracers which differ considerably from long-lived tracers. Repeating these

tests on $CO_2$ and $CH_4$, we found that using $\beta = 2$ is working more effectively, i.e. the weights $w_{jk}$ become even smaller in smooth regions and larger in regions with mass gradients. As this is an even number, $sgn\left(\phi_{jk}^* - \phi_{jk}^L\right)$ needs to be considered in (4) to allow preservation of monotonicity and positive definiteness. Moreover, to avoid erroneously large corrections in the stratosphere, the weight $w_{jk}$ is scaled by a factor $\frac{p_{jk}}{p_{j0}}$ that reflects the density variation from the surface to the top of the atmosphere. Since IFS uses a pressure based vertical coordinate, a good option is the ratio of the pressure at grid-point $jk$ ($p_{jk}$)

to the surface pressure below this grid-point ($p_{j0}$).

## 3    Experiments

Several $CO_2$ and $CH_4$ simulations using the IFS have been performed to test the influence of the global tracer mass fixers on their inter-hemispheric gradient. The global proportional fixer has been used for the low resolution simulations and shown to provide statisfactory results in terms of gradients in the $CO_2$ simulation (Agusti-Panareda et al., 2014) and $CH_4$ in the

TRANSCOM model intercomparison studies (Saito et al., 2013; Locatelli et al., 2013). However, it is not clear whether this is still the case for the high resolution simulations. For this reason, the global proportional fixer is compared with the Bermejo and Conde (2002) using two different resolutions. A low resolution corresponding to approximately 80 km in the horizontal with 60 model levels, i.e. the same as the one used by the ECMWF ERA-Interim re-analysis and similar to that used in climate simulations (e.g. Collins et al., 2011). The other resolution is approximately 16 km in the horizontal and 137 model levels, i.e.



**Table 1.** List of simulations at different resolutions and with different mass fixers performed from 1 March 2013 to 30 April 2014.

| Experiment description | Model grid resolution | Advection time step [s] |
|---|---|---|
| High resolution without fixer | TL1279, L137 | 600 |
| High resolution with proportional fixer | TL1279, L137 | 600 |
| High resolution with Bermejo & Conde (B&C) fixer | TL1279, , L137 | 600 |
| Low resolution without fixer | TL255, L60 | 2700 |
| Low resolution with proportional fixer | TL255, L60 | 2700 |
| Low resolution with Bermejo & Conde (B&C) fixer | TL255, L60 | 2700 |

following the operational NWP resolution also used in the operational $CO_2$ and $CH_4$ CAMS forecasts. The model time steps depend on the model resolution, corresponding to 10 and 45 minutes for high and low resolutions respectively. A list of all the experiments can be found in Tab1.

The simulations are performed using the cyclic forecast configuration with the IFS NWP model. This means that the mete-
orology is re-initialized at 00 UTC using the operational ECMWF NWP analysis, but the $CO_2$ and $CH_4$ tracers are allowed to evolve freely, i.e. without any constraint from observations. The transport in the IFS is based on the semi-lagrangian advection scheme (Temperton et al., 2001; Hortal, 2002; Untch and Hortal, 2006) described in the previous section, as well as a turbulent mixing scheme (Beljaars and Viterbo, 1998; Koehler et al., 2011; Sandu et al., 2013) and a convection scheme (Tiedtke, 1989; Bechtold et al., 2008, 2014).

The $CH_4$ fluxes and chemical sink in the simulations are based on prescribed climatologies and inventories as used by the operational CAMS $CH_4$ analysis and forecast (see Massart et al., 2014) following the prior fluxes and chemical sink of Bergamaschi et al. (2009) flux inversion system, except for the fire emissions from the GFAS dataset (Kaiser et al., 2012). The surface fluxes of $CO_2$ are also the same as used in the operational $CO_2$ analysis and forecast (see Agusti-Panareda et al., 2014, for a detailed description). They are all prescribed from inventories and climatologies, except for the land biogenic $CO_2$ fluxes which are modelled online by the CTESSEL Carbon module (Boussetta et al., 2013). A flux adjustment scheme has been implemented to correct for biases in the NEE budget with respect to a climatology of optimized fluxes from Chevallier et al. (2010) (see Agustí-Panareda et al., 2016, for further details). The fluxes for the high and low resolution are based on the same inventories and model. The global budget for the prescribed fluxes is the same, but their resolution is different. Because of that the gradients are sharper in the high resolution as the emission hotspots are characterized by stronger fluxes with the same mass distributed over a smaller area. For the modelled fluxes, the climate drivers such as radiation, soil moisture and temperature might vary with the resolution, and therefore the fluxes will not necessarily be the same. This only affects $CO_2$ as $CH_4$ only has prescribed fluxes.

The $CO_2$ and $CH_4$ simulations have been performed from 1st March 2013 to 30 April 2014. The aim is to test the annual accumulation of the error associated with mass conservation and the impact of the implemented mass fixer. The last month from 7th of March to 10th of April was used to compare with the observations from the Polarstern cruise (Klappenbach et al.,



2015) providing a north south transect across the Atlantic of total column averaged $CO_2$ and $CH_4$, together with observations from the Total Carbon Column Observing Network (TCCON) network (Wunch et al., 2011). A description of the observations used to assess the experiments is given in the next section.

## 4  Observations

The ship-based Polarstern dataset (Klappenbach et al., 2015) provides an excellent opportunity to assess the inter-hemispheric gradient, as it samples mainly oceanic well-mixed background air. The research vessel "Polarstern" took off from Cape Town (34°S,18°E), South Africa, on March 5, 2014, and entered port at Bremerhaven (54°N, 19°E), Germany, on April 14, 2014. During the cruise, an EM27/SUN near-infra-red-spectrometer was deployed onboard Polarstern. It collected direct-sun absorption spectra allowing to retrieve $XCO_2$ and $XCH_4$ with high precision and accuracy (Gisi et al., 2012; Hase et al., 2015;

Frey et al., 2015) as detailed for the Polarstern campaign by Klappenbach et al. (2015). Post-campaign deployment of the EM27/SUN side-by-side the TCCON spectrometer at Karlsruhe, Germany, allowed to calibrate $XCO_2$ and $XCH_4$ to the World Meteorological Organization (WMO) standard. Klappenbach et al. (2015) estimated the precision of the retrieved mole fractions to be to better than $0.2\,\mathrm{ppm}$ and $0.7\,\mathrm{ppb}$ for $XCO_2$ and $XCH_4$, respectively. This remote sensing technique samples the entire total column abundance and it is less dependent on localised sources in comparison to in-situ measurements.

All observations from $40^o$S to $40^o$N across the eastern Atlantic ocean were used. Information of the prior and averaging kernel were also used in order to be able to compare the observations with the model following Rodgers and Connor (2003).

While Polastern data provides a clear sampling of the meridional profile of background air representative of the large-scale inter-hemispheric gradient, it is not part of an operational network. For this reason, the evaluation of the inter-hemispheric gradient is corroborated using the TCCON observations. Observations from the TCCON network (Wunch et al., 2011) are

regularly used as a reference of total column $CO_2$ and $CH_4$ to calibrate and evaluate $CO_2$ and $CH_4$ products by the satellite community (e.g. Butz et al., 2011; Oshchepkov et al., 2013) and modelling community (Keppel-Aleks et al., 2011; Saito et al., 2012; Agusti-Panareda et al., 2014; Massart et al., 2015, e.g.). In this study, we used the version GGG2014 of the TCCON data (tccon.ornl.gov). TCCON sites used to assess the inter-hemispheric gradient are listed in Table 2.

## 5  Results

The impact of the mass fixers is assessed with global budget diagnostics (sec. 5.1), monthly mean total column maps (sec. 5.2) and comparisons with observations of the inter-hemispheric gradient (sec. 5.3).

For the global mass diagnostics, the mass of the $CO_2$ and $CH_4$ tracers is computed using Eq. 1. In the results that follow, the global error in tracer mass conservation during the advection to be corrected is computed as molar fraction in $ppm$ following:

$$DM = \frac{M(\phi^*, p^*) - M(\phi^o, p^o)}{M(p^o)} \frac{m_{air}}{m_{co2}} \times 10^6 \qquad (5)$$



**Table 2.** List of the TCCON stations used in this study and ordered by latitude from north to south.

| Site | Lat | Lon | Reference |
|------|-----|-----|-----------|
| Eureka | 80.05 | -86.42 | Strong et al. (2014) |
| Sodankylä | 67.37 | 26.63 | Kivi et al. (2014) |
| Karlsruhe | 49.10 | 8.44 | Hase et al. (2014) |
| Garmisch | 47.48 | 11.06 | Sussmann and Rettinger (2014) |
| Park Falls | 45.94 | -90.27 | Wennberg et al. (2014a) |
| Rikubetsu | 43.46 | -143.77 | Morino et al. |
| Lamont | 36.60 | -97.49 | Wennberg et al. (2014b) |
| Izaña | 28.30 | -16.48 | Blumenstock et al. (2014) |
| Ascension Island | -7.92 | -14.33 | Feist et al. (2014) |
| Darwin | -12.43 | 130.89 | Griffith et al. (2014a) |
| Wollongong | -34.41 | 150.88 | Griffith et al. (2014b) |
| Lauder 125HR | -45.05 | 169.68 | Sherlock et al. (2014) |

where $p^*$ is the pressure field after advection which has been corrected with a mass fixer to conserve global atmospheric mass (i.e. $M(p^o) = M(p^*)$).

### 5.1 Global mass conservation error

The instantaneous global mean mass conservation error per time step computed for the low and high resolution simulations using Eq. 5 is mostly positive (Fig. 1). The value oscillates around $1.2 \times 10^{-4}$ ppm for $CO_2$ and around $2.6 \times 10^{-3}$ ppb for $CH_4$ in the low resolution simulation. The error in the high resolution simulation is only slightly lower for $CO_2$ ($0.8 \times 10^{-4}$ ppm) and much lower for $CH_4$ ($0.6 \times 10^{-3}$ ppb) than in the low resolution simulation. The oscillations around the mean value are also smaller.

Although the instantaneous global mass conservation error per time step is small relative to the mean value of $CO_2$ and $CH_4$ (400 ppm and 1800 ppb respectively), the error is accumulated during the simulation. If the simulation is not re-initialized but cycled from one day to the next as in cyclic forecasts Agusti-Panareda et al. (2014) or climate runs, then this error will grow with time as shown in Fig. 2. The error growth rate is faster in the high resolution than in the low resolution simulation by a factor of 3.2 for $CO_2$ and 1.1 for $CH_4$, despite the smaller instantaneous errors in the high resolution simulation. This is because the time step is a factor 4.5 smaller than in the low resolution simulation. Therefore, the advection scheme is called more frequently, leading to a faster error accumulation. After one month, the conservation error reaches the value of 0.37 ppm for $CO_2$ and 2.79 ppb for $CH_4$ in the high resolution simulation. This is equivalent to an annual growth of 4.4 ppm/year and

33.0 ppb/year for $CO_2$ and $CH_4$ respectively. These error values are larger than the current observed growth of $CO_2$ (from 1 to 3 ppm/year; see Le Quéré et al., 2014) and $CH_4$ (from 0.6 to 16 ppb/year; see Dlugokencky et al., 2009; Kirschke et al., 2013).

## 5.2 Impact of mass fixers on total column $CO_2$ and $CH_4$ spatial distribution

The maps of mean $XCO_2$ and $XCH_4$ from 7 March to 10 April 2014 during the period of the Polarstern cruise (Figs 3 and 4) highlight the dominant inter-hemispheric gradient. After approximately one year of simulation without the mass fixer, the mean values of $XCO_2$ and $XCH_4$ are much higher everywhere, but particularly in the source regions in the northern hemisphere, e.g. over southeast Asia. The high resolution simulation in Figs 4(a) and 3(a) displays an enhanced increase with respect to the low resolution simulation (Figs 3(b) and 4(b)). For example, in southeast Asia the $XCO_2$ enhancement is around 4 ppm and

the $XCH_4$ enhancement is around 40 ppb.

Both proportional and Bermejo & Conde mass fixers reduce the mean $XCO_2$ and $XCH_4$ values everywhere, as intended. However, the proportional mass fixer leads to slightly different spatial distribution for the high and low resolution simulations (Figs 3(c,d) and 4(c,d)). Whereas the two spatial distributions obtained by using Bermejo & Conde remain closer to one another for the two different resolutions (Figs 3(e,f) and 4(e,f)). Some differences in the regions of sources and sinks are expected since

the surface fluxes are also affected by the resolution change, e.g. emission hotspots can be distributed over a smaller area and become more intense. However, this is not the case over Antarctica and the southern ocean where surface fluxes are very weak. The impact of the resolution south of 40$^o$S is indeed striking, particularly for the proportional mass fixer (Figs 3(c)(d) and 4(c)(d)). Over that region the mean $XCO_2$ and $XCH_4$ is 2 to 4 ppm and 20 to 40 ppb lower in the proportional mass fixer simulation at high resolution than all the other simulations. This large-scale mean negative difference cannot be explained by

differences in fluxes nor transport. Thus, it has to be linked to the mass conservation error and the effect of the proportional mass fixer, enhanced by the action of the mass fixer at high resolution (see sec. 5.1).

The effect of the mass fixers can be seen more clearly in Figs. 5 and 6 by computing the difference between the fields resulting from the different mass fixers with the fields from the simulation without any mass fixer. The proportional mass fixer removes mass quite uniformly for both the high and low resolution simulations, albeit with higher magnitude for the

high resolution case (Figs 5(a)(b) and 6(a)(b)). For example, the decrease in $XCO_2$ is around 2 ppm in the low resolution simulation, and around 10 ppm in the high resolution simulation. The $XCH_4$ decrease is not as uniform as in $XCO_2$, being larger in the northern hemisphere mid-latitudes by approximately 10 ppb at high resolution. On the other hand, the Bermejo & Conde mass fixer removes even more mass in the northern hemisphere than in the southern hemisphere, particulary at high resolution (see Figs 5(c)(d) and 6(c)(d)). This is a desirable effect, since the conservation error is expected to be larger closer

to the sources/sinks in the northern hemisphere.



### 5.3 Evaluation of inter-hemispheric gradient with observations

Comparing the simulations to the observed north-south transect in March/April 2014 we see that all the model simulations can represent the sign of the $XCO_2$ and $XCH_4$ gradient with larger values in the northern hemisphere and lower in the southern hemisphere (see Figs 7 and 8).

The errors with respect to both TCCON and Polarstern observed gradients are shown in Figs 9 and 10. The gradient of both $XCO_2$ and $XCH_4$ is steepest at high resolution without the mass fixer, compared to the lower resolution simulation and also to other simulations with mass fixer. This corroborates the detrimental enhancement $XCO_2$ and $XCH_4$ – particularly in the northern hemisphere – associated with the accumulation of mass conservation errors. The proportional mass fixer also results in a gradient which is too steep, particularly at high resolution (see light blue line in Figs 7 and 8). The simulation with the

Bermejo & Conde fixer has the gradient closest to the observed profiles. It also presents the best consistency (i.e. smallest difference) between high and low resolution simulations.

The inter-hemispheric gradient can be quantified as the difference between the tracer in the northern hemisphere and southern hemisphere. Here we take between $20^oN$ and $50^oN$ and between $20^oS$ and $40^oS$ for the two hemispheres due to the availability of observations. For $XCO_2$ the observed difference is 4.29 ppm and 5.76 ppm using the Polarstern or the TCCON datasets

respectively. For $XCH_4$ the gradient is 53.81 ppb and 52.64 ppb for the same datasets respectively. The gradient for the different experiments is shown in Tables 3 to 6. All the low resolution simulations have a similar gradient of $XCO_2$ of approximately 7 ppm with a range of 0.7 ppm (Polarstern) and 0.6 ppm (TCCON). That is, the range of inter-hemispheric gradients at the low resolution is around 10 % of its value. Whereas the high resolution simulations have a larger range of 2 ppm corresponding to a 30% spread. This highlights the distorting effect of the mass conservation error on the inter-hemispheric gradient. For $XCH_4$

the effect is similar, albeit even more pronounced than for $XCO_2$ in the low resolution simulations, where the range of the inter-hemispheric gradient values is around 18 ppb (i.e. 34% of its value). At high resolution the $XCH_4$ range is around 34 ppb (i.e. 63%).

When looking at the impact of each fixer, we see that the simulation with the proportional mass fixer has the same error in inter-hemispheric gradient as the simulation without mass fixer (i.e. 4.3 to 5.9 ppm at high resolution and 1.6 to 3.4 ppm at low

resolution, comprising 75% to 140% of the error at high resolution and 32% to 79% at low resolution). It is clear that the error grows with high resolution. This against all expectations as the objective for high resolution simulations is to achieve a better accuracy. On the other hand, the Bermejo & Conde fixer is able to keep a closer gradient between the low and high resolution simulations (within 1 ppm and 2 ppb for $XCO_2$ and $XCH_4$). The resulting error with respect to both Polarstern and TCCON is nearly half the inter-hemispheric error of the proportional mass fixer.

These results are consistent with the station-to-station bias, which is computed as the standard deviation of the biases from the individual stations or cruise observations. The results are very similar when either there is no mass fixer or the proportional fixer mass is used. For $XCO_2$ the inter-station bias is 2 ppm and 1.2 ppm at high and low resolutions respectively. While for $XCH_4$ the inter-station bias ranges from 14 to 19 ppb and from 9 to 14 ppb at high and low resolutions respectively. The Bermejo & Conde is again showing an improvement with similar values for the high and low resolution simulations of around





1.4 ppm for $XCO_2$ and around 4.8 ppb for $XCH_4$. These values are in line with the variability of the bias in space and time obtained from satellite retrievals of GOSAT (Dils et al., 2014).

The effect of both proportional and Bermejo & Conde mass fixers on the bias with respect to observations is similar. They both manage to reduce the bias from around 2% to less than 0.4% for $XCO_2$ and from around 4% to less than 1% for $XCH_4$. It is worth noting that even for the bias, the Bermejo & Conde is able to have a reduction of the bias error of at least 0.1% with respect to the proportional mass fixer, leading to an overall bias of 0.2% ($\sim 0.7$ ppm).

It is also remarkable that the resulting errors associated with the inter-hemispheric gradient are the same when using TCCON and Polarstern observations, despite being at different sampling sites (i.e. along different longitudes). The uniformity of the results throughout the globe means that the main error source is global. This is consistent with global error source of the mass fixer. Therefore, it strengthens the suggestion that the observations used here are able to detect the effects of the mass fixer more than the other effects associated with localized error sources from local fluxes and/or regional transport.

## 6   Conclusions

Atmospheric transport schemes used in models to monitor/predict climate change and atmospheric composition are required to conserve the global mass of atmospheric tracers. Thus, the use of numerical methods that do not inherently conserve mass, such as the widely used semi-lagrangian advection scheme, entail the application of mass fixers to ensure the preservation of the global mass. This is particularly important for long-lived greenhouse gases for which the interesting signals to monitor (e.g. annual growth rates and large-scale spatial gradients) are weak compared to their background values. This paper explores the impact of two global mass fixers on the inter-hemispheric gradient of total column averaged $CO_2$ and $CH_4$ using observations from the Polarstern cruise and the TCCON network. The widely used proportional fixer is compared to the Bermejo & Conde fixer, presenting a feasible alternative in the context of operational atmospheric transport models.

Two different resolutions are also compared, the first one is a typical climate resolution of 80 km and 60 model levels and the second one is the current resolution used in NWP at 16 km in the horizontal and 137 model levels. Results show clearly that errors accumulate much faster for the high resolution simulations and after one year the mass conservation error exceeds by far the observed annual growth rate of $CO_2$ and $CH_4$. The mass conservation errors of $XCO_2$ and $XCH_4$ grow faster in the nothern hemisphere than in the southern hemisphere, causing a steepening of the inter-hemispheric gradient. The proportional mass fixer applies a uniform correction globally because it only depends on the background value which is uniformly high. Thus, the proportional fixer is efficient at removing the global bias, but it cannot correct for the steepening of the inter-hemispheric gradient. This is detected as an artificial reduction of $XCO_2$ and $XCH_4$ in the southern hemisphere and a resulting excess in the northern hemisphere when comparing with observations as depicted in Fig. 11. On the other hand, the alternative Bermejo & Conde fixer enhances the mass correction in the regions where gradients are steeper. $CO_2$ and $CH_4$ gradients are steeper where their surface fluxes are stronger, i.e. in the northern hemisphere. The Bermejo & Conde mass fixer correction is therefore latitudinally dependent and it is able to correct the inter-hemispheric gradient, bringing the low and high resolution simulations closer to each other and closer to the observations.





In summary, the tests performed using the IFS show that although the proportional mass fixer is suitable at low resolutions currently used in NWP re-analysis and climate simulations, it is not suitable for NWP resolutions at 16 km and 137 vertical levels. An alternative global mass fixer based on Bermejo & Conde has been shown to work reasonably well when compared to observations at both low and high resolutions without too much additional complexity or cost.

**Code and/or data availability**

The C-IFS source code is integrated into ECWMF's IFS code, which is only available subject to a licence agreement with ECMWF. ECMWF member-state weather services and their approved partners will get access granted. The IFS code without modules for assimilation and chemistry can be obtained for educational and academic purposes as part of the openIFS release (https:// software.ecmwf.int/wiki/display/OIFS/OpenIFS+Home). A detailed documentation of the IFS code is available from

https://software.ecmwf.int/wiki/display/IFS/CY40R1+ Official+IFS+Documentation. The output from C-IFS can be requested via http://copernicus-support.ecmwf.int. The Polarstern data is available in the Supplement of Klappenbach et al. (2015) at doi:10.5194/amt-8-5023-2015-supplement. The TCCON data (version GGG2014 ) is available from tccon.ornl.gov.

*Acknowledgements.* This study has been funded by the European Commission under Monitoring of Atmospheric Composition and Climate project and the Copernicus Atmosphere Monitoring Service.

FK and AB acknowledge support by Frank Hase, KIT, for instrument development and data reduction, by the Emmy-Noether program of the Deutsche Forschungsgemeinschaft (DFG) through grant BU2599/1-1 (RemoteC), and by Alfred Wegener Institute (AWI), Helmholtz Centre for Polar and Marine Research, for operating RV Polarstern and granting access to its infrastructures.

TCCON data were obtained from the TCCON Data Archive, hosted by the Carbon Dioxide Information Analysis Center (CDIAC) - tccon.ornl.gov. The authors would like to acknowledge the PIs of the different TCCON stations used in this study: Kimberly Strong (Eureka,

Canada) Rigel Kivi (Sodankylä, Finland), Frank Hase (Karlsruhe, Germany), Ralf Sussmann (Garmisch, Germany), Paul Wennberg (Park Falls, Lamont, USA), Matthias Schneider (Izaña, Spain), Dietrich Feist (Ascension Island), David Griffith (Darwin, Wollongong, Australia), Dave Pollard and Vanessa Sherlock (Lauder, New Zealand).The operation at the Rikubetsu TCCON site is supported in part by the budget from the GOSAT data validation project funded by the Ministry of Environment, Japan.

The authors are grateful to Sebastien Massart and Johannes Flemming for useful discussions and comments during the completion of this

work.



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

**Table 3.** $XCO_2$ inter-hemispheric gradient (IHG) error [MODEL - OBS] statistics for simulations with different resolution and different mass fixers with respect to observations from the Polarstern cruise.

| Data | IHG [ppm] | IHG error [ppm] | Overall bias [ppm] (%) | Inter-station bias [ppm] (%) |
|---|---|---|---|---|
| OBS | 4.29 | | | |
| Low resolution without fixer | 7.81 | 3.52 | 2.70 (0.68) | 1.54 (0.39) |
| Low resolution with proportional fixer | 7.70 | 3.42 | 0.82 (0.21) | 1.50 (0.38) |
| Low resolution with B&C | 7.11 | 2.82 | 0.62 (0.16) | 1.30 (0.33) |
| High resolution without fixer | 10.54 | 6.25 | 7.86 (1.98) | 2.54 (0.64) |
| High resolution with proportional fixer | 10.17 | 5.89 | 1.36 (0.34) | 2.40 (0.60) |
| High resolution with B&C | 7.97 | 3.69 | 0.69 (0.17) | 1.61 (0.40) |
| Spread of low resolution simulations | 0.70 | 0.70 | 2.01 (0.51) | 0.24 (0.06) |
| Spread of high resolution simulations | 2.57 | 2.56 | 7.17 (1.81) | 0.93 (0.24) |
| Spread of low resolution B&C and proportional | 0.59 | 0.60 | 0.20 (0.04) | 0.20 (0.05) |
| Spread of high resolution B&C and proportional | 2.20 | 2.20 | 0.67 (0.17) | 0.79 (0.20) |





**Table 4.** XCO$_2$ inter-hemispheric gradient (IHG) error [MODEL - OBS] statistics for simulations with different resolution and different mass fixers with respect to observations from TCCON.

| Data | IHG [ppm] | IHG error [ppm] | Overall bias [ppm] (%) | Inter-station bias [ppm] (%) |
|---|---|---|---|---|
| OBS | 5.76 | | | |
| Low resolution without fixer | 7.48 | 1.71 | 2.71 (0.68) | 1.21 (0.30) |
| Low resolution with proportional fixer | 7.45 | 1.68 | 0.83 (0.21) | 1.20 (0.30) |
| Low resolution with B&C | 6.93 | 1.16 | 0.69 (0.17) | 1.02 (0.26) |
| High resolution without fixer | 10.14 | 4.38 | 7.94 (1.99) | 2.16 (0.54) |
| High resolution with proportional fixer | 10.04 | 4.28 | 1.44 (0.36) | 2.13 (0.54) |
| High resolution with B&C | 8.10 | 2.34 | 0.88 (0.22) | 1.45 (0.37) |
| Spread of low resolution simulations | 0.55 | 0.55 | 2.02 (0.51) | 0.19 (0.05) |
| Spread of high resolution simulations | 2.04 | 2.04 | 7.06 (1.77) | 0.71 (0.17) |
| Spread of low resolution B&C and proportional | 0.52 | 0.52 | 0.14 (0.04) | 0.18 (0.05) |
| Spread of high resolution B&C and proportional | 1.94 | 1.94 | 0.56 (0.14) | 0.68 (0.17) |

**Table 5.** XCH$_4$ inter-hemispheric gradient (IHG) error [MODEL - OBS] statistics for simulations with different resolution and different mass fixers with respect to observations from the Polarstern cruise.

| Data | IHG [ppb] | IHG error [ppb] | Overall bias [ppb] (%) | Inter-station bias [ppb] (%) |
|---|---|---|---|---|
| OBS | 53.81 | | | |
| Low resolution without fixer | 73.42 | 19.61 | 41.09 (2.28) | 9.88 (0.55) |
| Low resolution with proportional fixer | 70.65 | 16.84 | 1.74 (0.10) | 8.91 (0.50) |
| Low resolution with B&C | 54.29 | 0.48 | 6.58 (0.37) | 4.84 (0.27) |
| High resolution without fixer | 92.00 | 38.19 | 55.83 (3.10) | 16.84 (0.94) |
| High resolution with proportional fixer | 88.19 | 34.38 | 6.05 (0.34) | 15.36 (0.85) |
| High resolution with B&C | 55.71 | 1.90 | 1.82 (0.10) | 4.64 (0.26) |
| Spread of low resolution simulations | 19.13 | 19.13 | 39.35 (2.18) | 5.05 (0.28) |
| Spread of high resolution simulations | 36.29 | 36.29 | 54.01 (3.00) | 12.20 (0.68) |
| Spread of low resolution B&C and proportional | 16.36 | 16.36 | 2.01 (0.11) | 4.07 (0.23) |
| Spread of high resolution B&C and proportional | 33.90 | 32.48 | 4.23 (0.24) | 10.72 (0.59) |




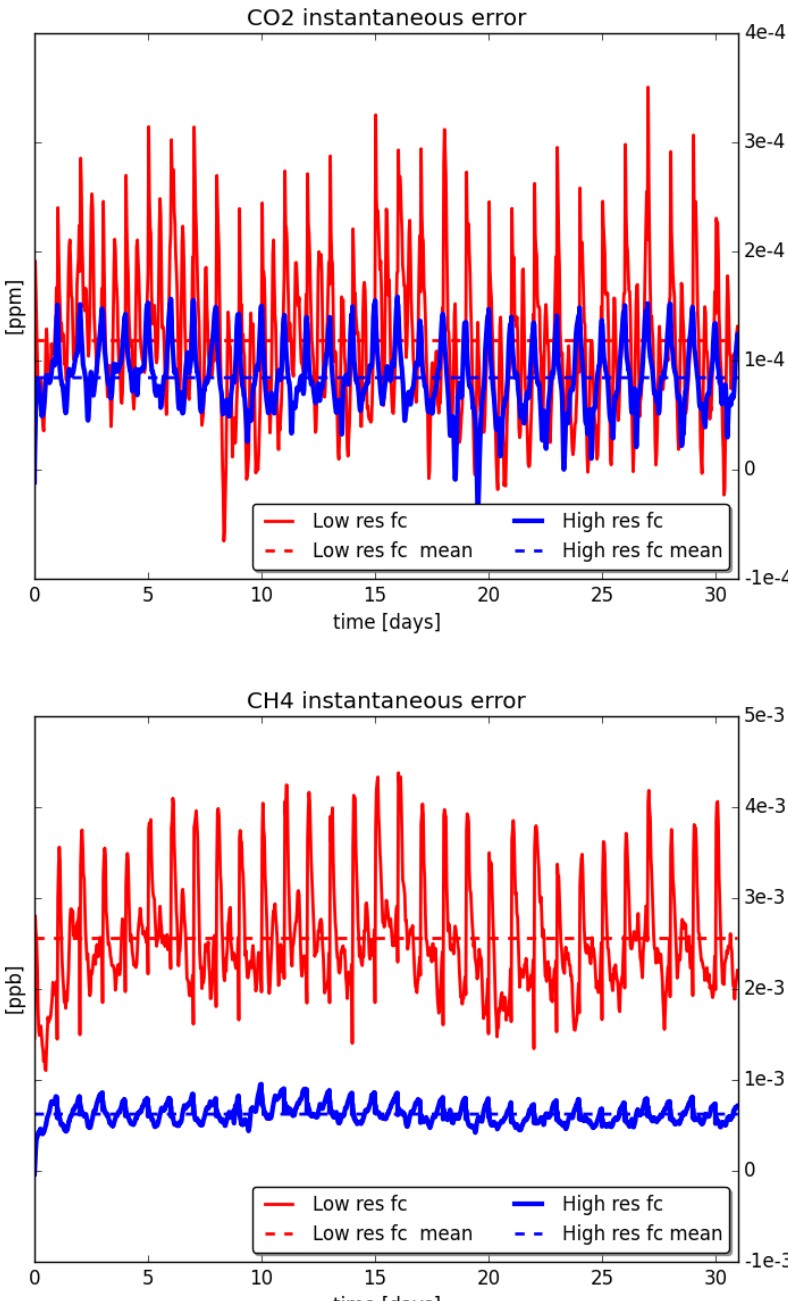

**Figure 1.** Instantaneous global mean mass conservation error for $CO_2$ [ppm] and $CH_4$ [ppb] from 1 to 31 March 2013. Low/high resolution experiments are depicted by red/blue lines.





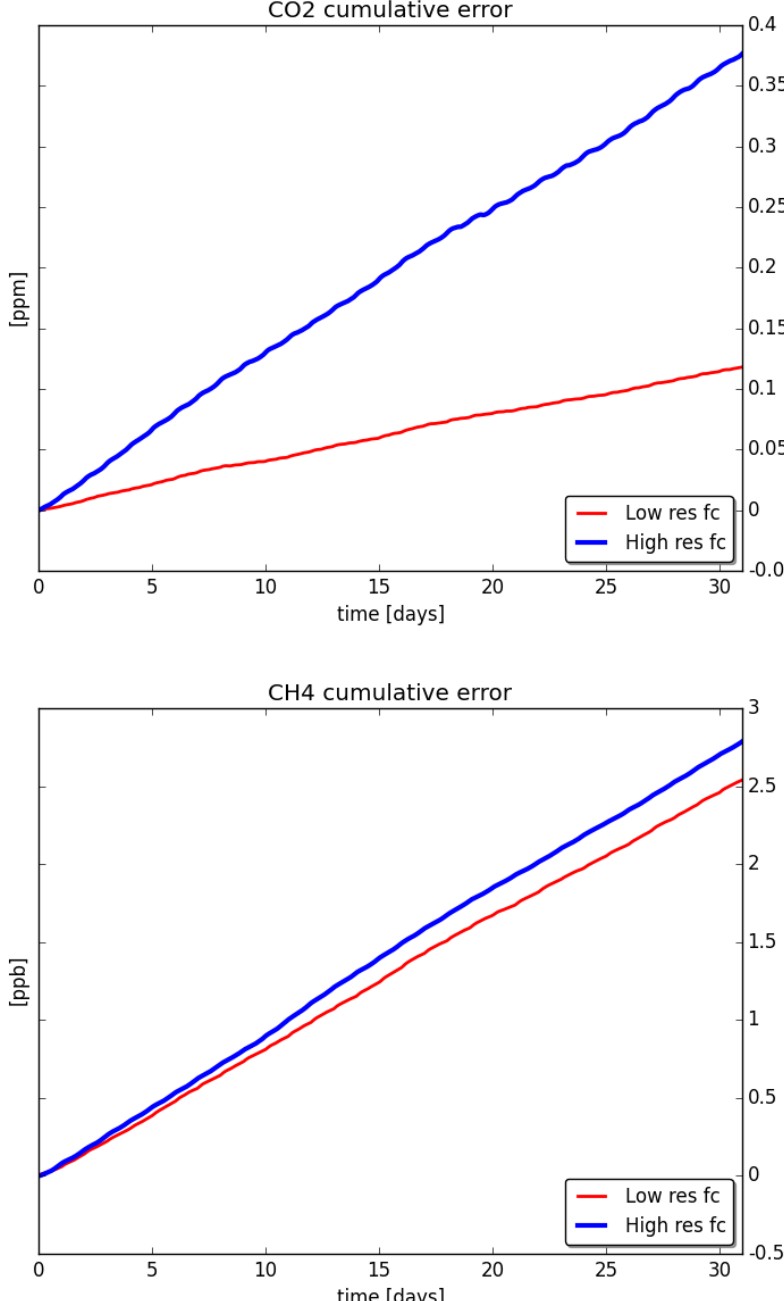

**Figure 2.** Cumulative global mean mass conservation error for $CO_2$ [ppm] and $CH_4$ [ppb] from 1 to 31 March 2013. Low/high resolution experiments are depicted by the red/blue lines.



**Figure 3.** Mean $XCO_2$ [ppm] from 7 March to 10 April 2014 for the high resolution (left panels) and low resolution (right panels) simulations. The effect of the different mass fixers is shown in the different rows. Details of the simulations can be found in Table 1. The pink and black triangles mark the location of the reference observations from TCCON and Polarstern cruise respectively.



**Figure 4.** Mean $XCH_4$ [ppb] from 7 March to 10 April 2014 for the high resolution (left panels) and low resolution (right panels) simulations. The effect of the different mass fixers is shown in the different rows. Details of the simulations can be found in Table 1. The pink and black triangles mark the location of the reference observations from TCCON and Polarstern cruise respectively.





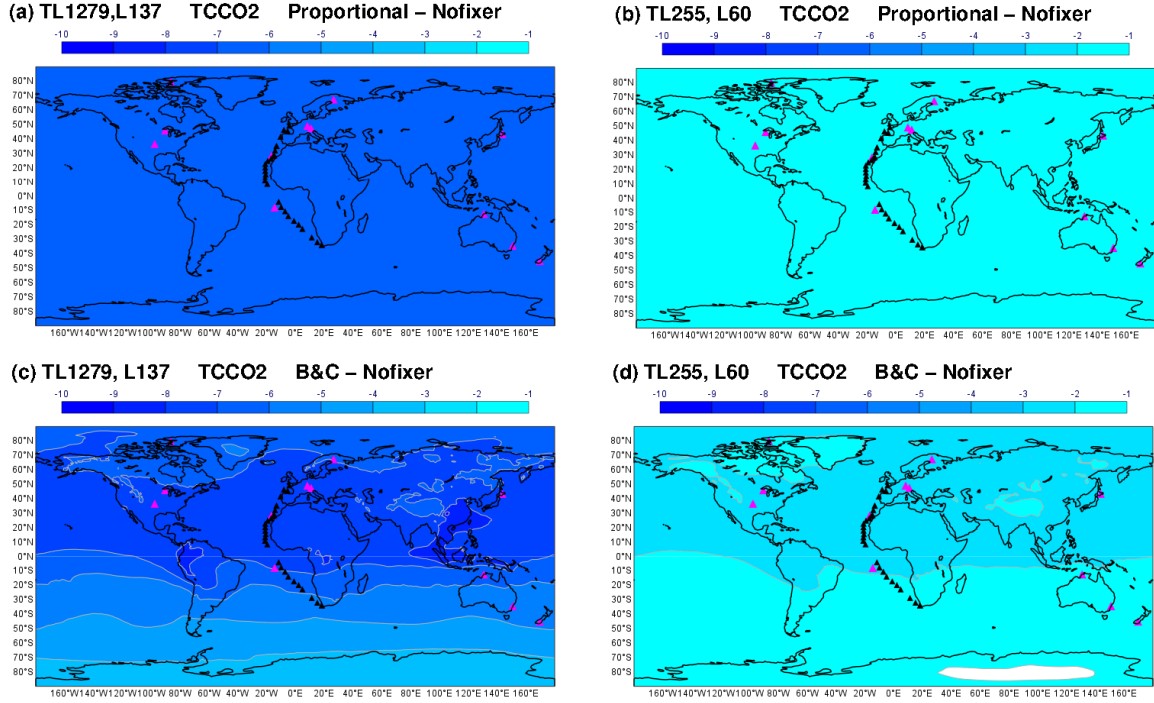

**Figure 5.** Difference in mean $XCO_2$ [ppm] between: (a,b) the simulations using the proportional mass fixer and the simulation without mass fixer at high and low resolution respectively; and (c,d) the simulation with Bermejo and Conde (B&C) and the simulation without mass fixer at high and low resolutions respectively. The period covered and the marking of the observation sites are the same as in Fig. 3.



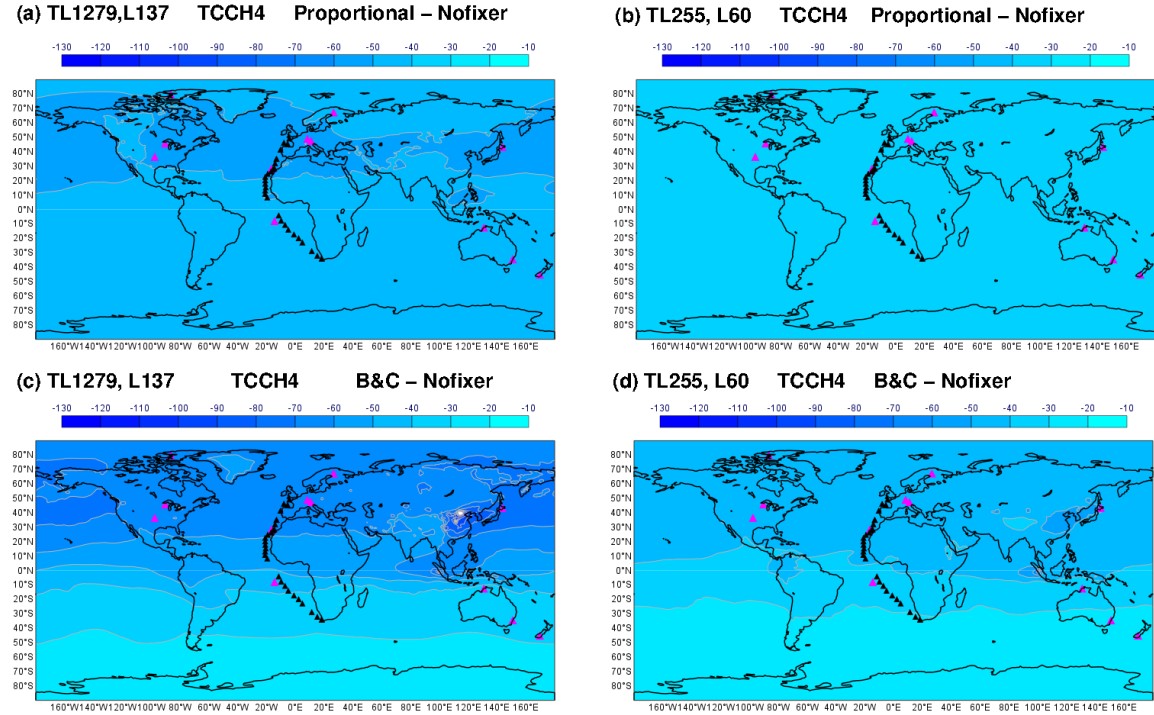

**Figure 6.** Difference in mean $XCH_4$ [ppb] between: (a,b) the simulations using the proportional mass fixer and the simulation without mass fixer at high and low resolution respectively; and (c,d) the simulation with Bermejo and Conde (B&C) and the simulation without mass fixer at high and low resolutions respectively. The period covered and the marking of the observation sites are the same as in Fig. 4.





**Figure 7.** (a) Map showing the daily mean sampling location of Polarstern cruise. (b,c) Comparisons of latitudinal distribution of $XCO_2$ and $XCH_4$ as derived from monthly mean (7 March to 10 April) Polarstern observations (black) and simulations using different mass fixers at different resolutions: red/orange without mass fixer at low/high resolutions blue/cyan with the proportional mass fixer at low/high resolutions; and green/light green with the Bermejo & Conde fixer and low/high resolutions respectively. See Table 1 for a more detailed description of the experiments.





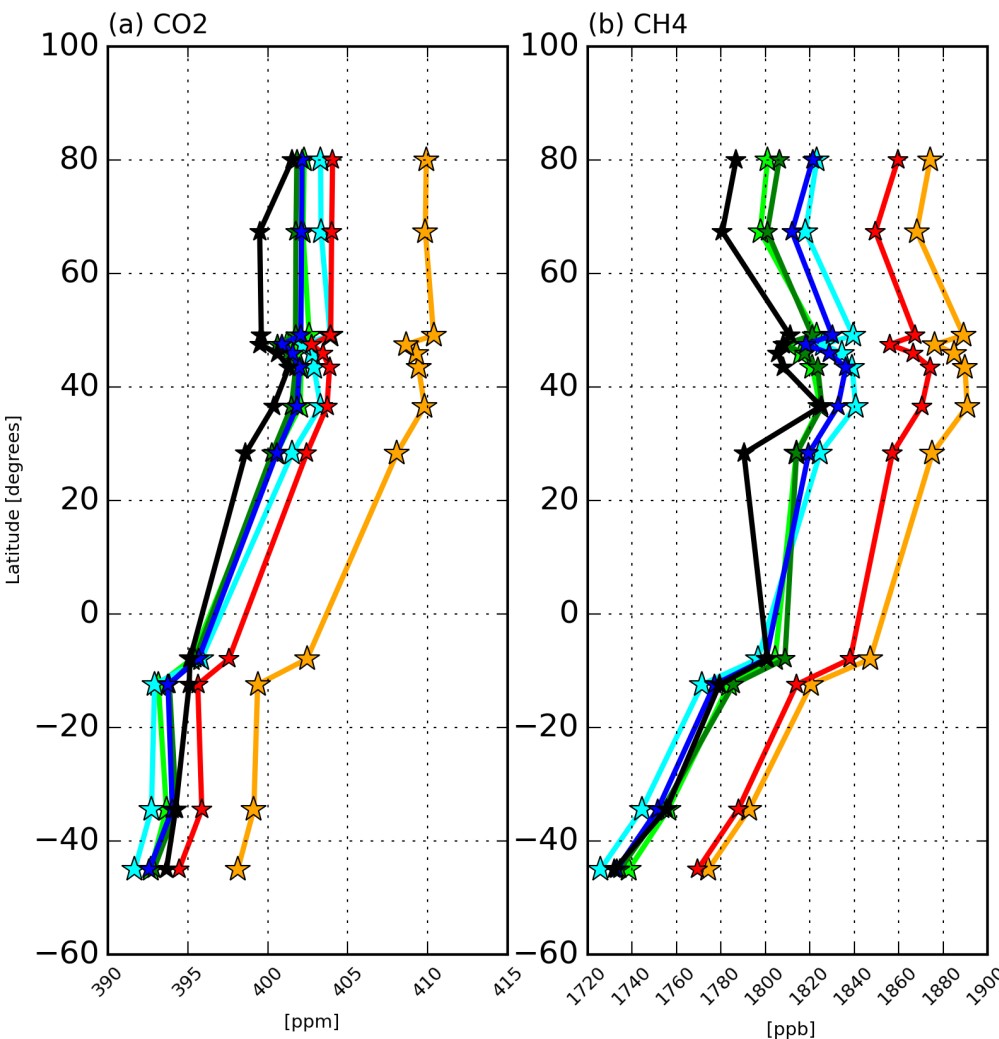

**Figure 8.** Comparisons of latitudinal distribution of (a) $XCO_2$ and (b) $XCH_4$ as derived from monthly mean (7 March to 10 April) TCCON sites (black, see Table 2)) and simulations using different mass fixers at different resolutions: red/orange without mass fixer at low/high resolutions blue/cyan with the proportional mass fixer at low/high resolutions; and green/light green with the Bermejo & Conde fixer and low/high resolutions respectively. See Table 1 for a more detailed description of the experiments.





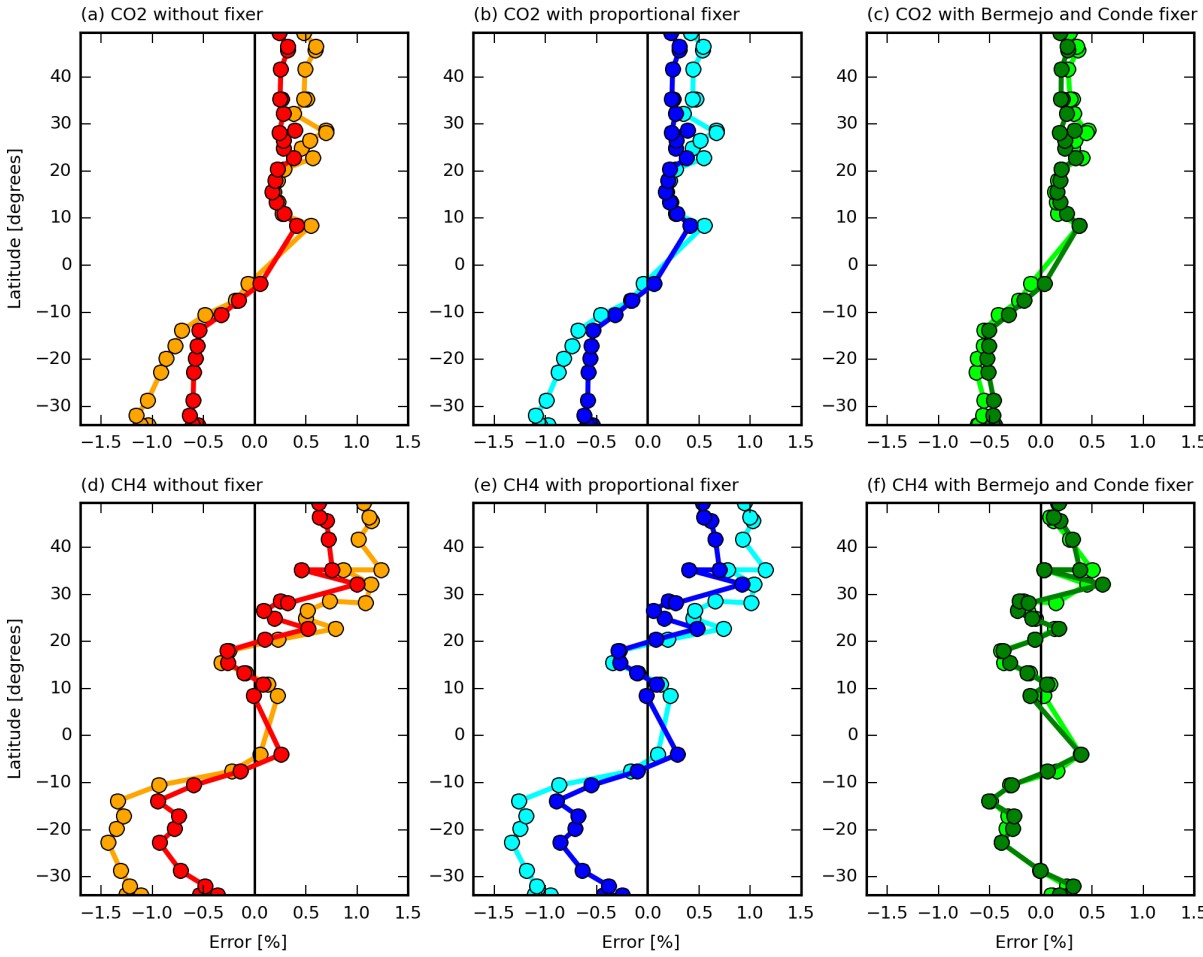

**Figure 9.** Error [%] of modelled latitudinal monthly mean (7 March to 10 April) distribution computed as (MODEL-OBS)/OBS using different tracer mass fixers and different resolutions for (a-c) $XCO_2$ and (d-f) $XCH_4$ with respect to the observed distribution from Polarstern. Dark/light colours correspond to the simulations at low/high resolution respectively.





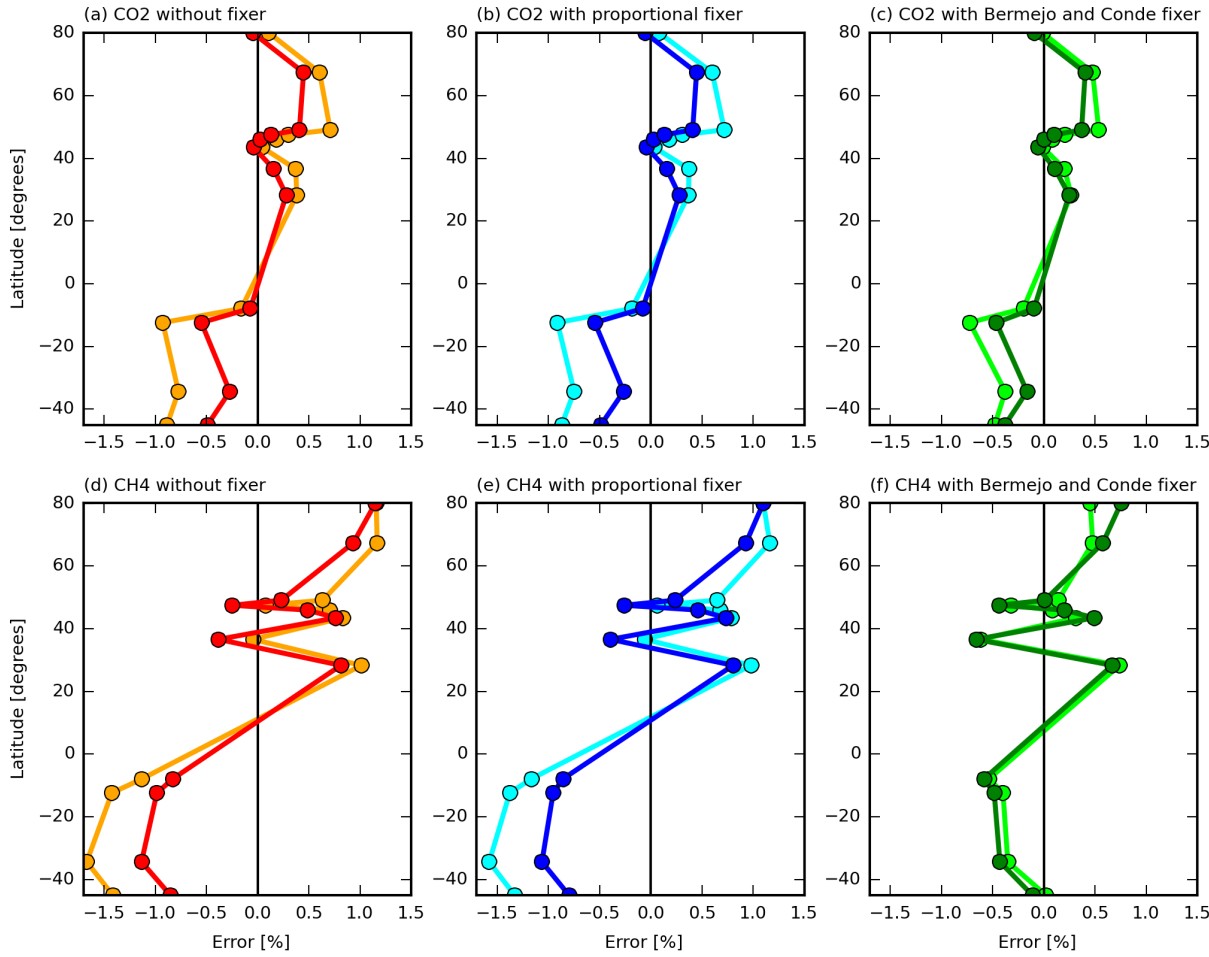

**Figure 10.** Error [%] of modelled latitudinal monthly mean (7 March to 10 April) distribution computed as (MODEL-OBS)/OBS using different tracer mass fixers and different resolutions for (a-c) $XCO_2$ and (d-f) $XCH_4$ with respect to the observed distribution from TCCON. Dark/light colours correspond to the simulations at low/high resolution respectively.





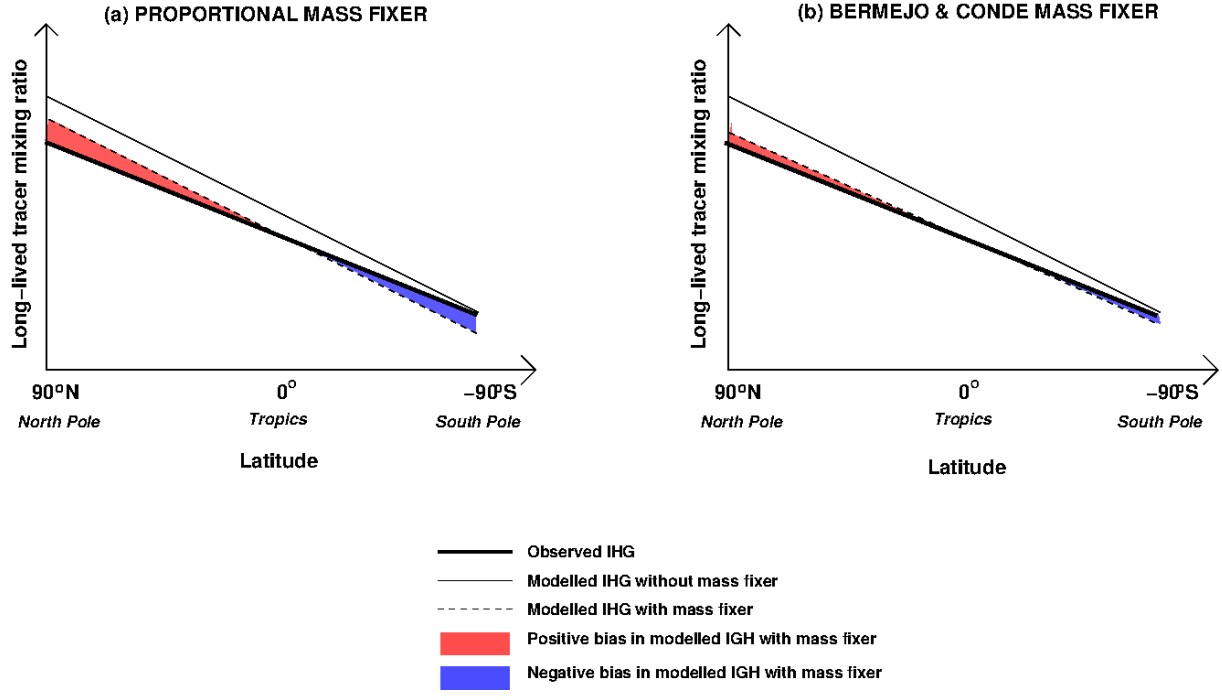

**Figure 11.** Schematic illustrating the impact of the (a) proportional and (b) Bermejo & Conde mass fixers on the inter-hemispheric gradient of $XCO_2$ and $XCH_4$. Note that the area between the dash line and thin solid line depicting the global correction of tracer mass should be the same for the two mass fixers.





**Table 6.** XCH$_4$ inter-hemispheric gradient (IHG) error [MODEL - OBS] statistics for simulations with different resolution and different mass fixers with respect to observations from TCCON.

| Data | IHG [ppb] | IHG error [ppb] | Overall bias [ppb] (%) | Inter-station bias [ppb] (%) |
|---|---|---|---|---|
| OBS | 52.64 | | | |
| Low resolution without fixer | 77.54 | 24.90 | 52.28 (2.92) | 14.17 (0.79) |
| Low resolution with proportional fixer | 76.06 | 23.42 | 14.77 (0.83) | 13.76 (0.77) |
| Low resolution with B&C | 60.96 | 8.32 | 11.47 (0.64) | 9.02 (0.50) |
| High resolution without fixer | 91.62 | 38.98 | 66.68 (3.72) | 18.76 (1.05) |
| High resolution with proportional fixer | 89.64 | 37.00 | 16.70 (0.93) | 18.16 (1.01) |
| High resolution with B&C | 59.78 | 7.14 | 9.90 (0.55) | 7.62 (0.43) |
| Spread of low resolution simulations | 16.58 | 16.58 | 40.81 (2.28) | 5.15 (0.29) |
| Spread of high resolution simulations | 31.84 | 31.84 | 56.78 (3.17) | 11.14 (0.62) |
| Spread of low resolution B&C and proportional | 15.10 | 15.10 | 3.30 (1.19) | 4.74 (0.27) |
| Spread of high resolution B&C and proportional | 29.86 | 29.86 | 6.80 (0.38) | 10.54 (0.58) |