# Peer review of "Improving the inter-hemispheric gradient of total column atmospheric $CO_2$ and $CH_4$ in simulations with the ECMWF semi-Lagrangian atmospheric global model"

_Geoscientific Model Development, 2016_

## Short Comment (SC1) · 20 Jul 2016

Dear authors,

In my role as Executive editor of GMD, I would like to bring to your attention our Editorial version 1.1:

http://www.geosci-model-dev.net/8/3487/2015/gmd-8-3487-2015.html

This highlights some requirements of papers published in GMD, which is also available on the GMD website in the 'Manuscript Types' section:

[Figure]

http://www.geoscientific-model-development.net/submission/manuscript_types.html

In particular, please note that for your paper, the following requirement has not been met in the Discussions paper:

- "The main paper must give the model name and version number (or other unique identifier) in the title."

Please add a version number of the ECMWF semi-Lagrangian atmospheric global model in the title upon your revised submission to GMD.

Yours,

Astrid Kerkweg

―――――――――――――――――――

---

## Referee Comment (RC1) · Anonymous Referee #1 · 29 Jul 2016

**Review of the manuscript:** Improving the inter-hemispheric gradient of total column atmospheric \chem{CO_2} and \chem{CH_4} in simulations with the ECMWF semi-Lagrangian atmospheric global model

A. Agusti-Panareda, M. Diamantakis, V. Bayona, F. Klappenbach, and A. Butz

The manuscript addresses the importtant issue of mass conservation of SL schemes in long time rank simulations.

The description of the numerical methods is clear and so is the presentation of the results. The numerical tests are relevant.

The discussion of the results is illuminating and helpful for other researchers.

The paper is excellent according to the principal criteria for acceptance, so my recommendation is that the paper should be accepted as is.

---

## Referee Comment (RC2) · Anonymous Referee #2 · 9 Aug 2016

The manuscript "Improving the inter-hemispheric gradient of total column atmospheric CO2 and CH4 in simulations with the ECMWF semi-Lagrangian atmospheric global model" by Anna Agusti-Panareda et al. describes a comparison of a taylored Bermejo and Conde mass fixer and the proportional mass fixer, the impact of the two mass fixers on the preservation of the CO2 and CH4 inter-hemispheric gradient, the influence of resolution on the mass conservation and performance of the mass fixers. The figures and tables are generally clear and well chosen. Overall the manuscript is clearly written, although there are technical corrections necessary to be fixed before publication. I recommend publication after these revisions have been carried out.

Comments:

CO2 and CH4 are long-lived greenhouse gases, thus I suggest considering a longer period (probably the whole simulation period: 1st March 2013 to 30 April 2014) for analysis (Fig. 1-6, 8, 10-11) excepting comparison with Polarstern observations. The main aim of the paper is improving the inter-hemispheric gradient of total column atmospheric CO2 and CH4, however it is useful to study vertical cross sections or/and profiles of mass conservation error.

Technical comments:

Page 3, line 9: performace -> performance

Figures 3-6: no labels for TCCON FTS sites; TCCO2, TCCH4 should be replaced with XCO2 and XCH4 respectively.

Figures 7-10: CO2, CH4 should be replaced with XCO2 and XCH4 respectively.

---

## Author Comment (AC1) · 6 Oct 2016

The authors would like to thank the reviewers for the comments which have been addressed below and have contributed to improve the clarity of the manuscript.

**General comments**

- *$CO_2$ and $CH_4$ are long-lived greenhouse gases, thus I suggest considering a*

*longer period (probably the whole simulation period: 1st of March 2013 to 30 April 2014) for analysis (Fig. 1-6, 10-11) excepting comparison with Polarstern observations.*

The mass conservation error is very small at the beginning of the simulation and it grows with time as it accumulates (see Fig. 1 for the accumulation throughout the whole period from 1st of March 2013 to 30 April 2014). For this reason, we have chosen to start the simulation one year before the Polarstern campaign in order to assess the accumulated errors after one year of simulation. If we include the whole simulation period in the evaluation of the mass fixer impact, then we would be looking a the mean impact which has the same pattern as the accumulated impact but a much smaller amplitude (see Fig. 2 for whole period compared Figs 5 and 6 in the manuscripts covering the Polarstern period). Thus, using the mean impact, instead of accumulated impact, would make the detection of significant differences between observations and simulations more difficult. This will be clarified in the revised manuscript (section 3) by emphasizing the importance of evaluating the accumulated impact of the mass fixer during the last month of the simulations, as opposed to the mean impact throughout the whole period.

• *The main aim of the paper is improving the inter-hemispheric gradient of total column atmospheric $CO_2$ and $CH_4$, however it is useful to study vertical cross sections and/or profiles of mass conservation error.*

In the proportional mass fixer experiments, the mass fixer correction is applied uniformly throughout the column, whereas in the taylored Bermejo and Conde mass fixer, most of the correction is performed at the lower levels where the atmospheric mass and the tracer mass are largest (Fig. 3). This is mentioned in section 2.2 of the manuscript. Further work should be done to compare the

impact of the mass fixer on the vertical gradients of $CO_2$ and $CH_4$ using observations, e.g. from aircrafts, but this is out of the scope of this study.

**Technical comments**

- *Page 3, line 9: performace -> performance* Done.

- *Figures 3-6: no labels for TCCON FTS sites; TCCO2, TCCH4 should be replaced with XCO2 and XCH4 respectively.* Done. We have not been able to include the TCCON labels on the map because they would obstruct the small plot. We have included a reference to the TCCON table listing all the sites and their lat/lon coordinates in the Figure captions instead.

- *Figures 7-10: CO2 and CH4 should be replaced with XCO2 and XCH4 respectively.* Done.
* * *
[Figure]

**(a)**

CO2 instantaneous error

[ppm]

time [days]

**(b)**

CO2 cumulative error

[ppm]

Low res fc
High res fc

time [days]

**(c)**

CH4 instantaneous error

[ppb]

time [days]

**(d)**

CH4 cumulative error

[ppb]

Low res fc
High res fc

time [days]

**Fig. 1.** Time series of instantaneous and accumulated global mean mass conservation error for CO2 and CH4 from 1 March 2013 to 30 April 2014. Low/high resolution experiments are depicted by red/blue lines.

**(a) TL1279, L137   XCO2     Proportional – Nofixer**

**(b) TL1279, L137 XCH4    Proportional – Nofixer**

**(c) TL1279, L137      XCO2      B&C – Nofixer**

**(d) TL1279, L137      XCH4      B&C – Nofixer**

**Fig. 2.** Difference in mean XCO2 [ppm] and XCH4 [ppb] from 1 March 2013 to 30 April 2014 between simulations using the proportional and no fixer (a,b) and B&C and no fixer (c,d) at high resolution.

**Fig. 3.** Vertical section of zonal mean differences in CO2 and CH4 from 7 March 2014 to 10 April 2014 between simulations with proportional and no fixer (a,b); and B&C and no fixer (c,d) at high resolution.

---

## Author Comment (AC2) · 6 Oct 2016

The authors would like to thank the reviewer for the positive and constructive comments on the manuscript.

---

## Author Response (AR1)

**Reply to referee 2 on manuscript entitled "Improving the inter-hemispheric gradient of total column atmospheric CO2 and CH4 in simulations with the ECMWF semi-Lagrangian atmospheric global model" by A. Agusti-Panareda et al.**

A.Agusti-Panareda, M. Diamantakis, V. Bayona, F. Klappenbach, A. Butz

October 28, 2016

The authors would like to thank the reviewers for the comments which have been addressed below and have contributed to improve the clarity of the manuscript. The modifications done to the revised manuscript have been highlighted in blue.

**General comments**

- *$CO_2$ and $CH_4$ are long-lived greenhouse gases, thus I suggest considering a longer period (probably the whole simulation period: 1st of March 2013 to 30 April 2014) for analysis (Fig. 1-6, 10-11) excepting comparison with Polarstern observations.*

  The mass conservation error is very small at the beginning of the simulation and it grows with time as it accumulates (see Fig. 1 for the accumulation throughout the whole period from 1st of March 2013 to 30 April 2014). For this reason, we have chosen to start the simulation one year before the Polarstern campaign in order to assess the accumulated errors after one year of simulation. If we include the whole simulation period in the evaluation of the mass fixer impact, then we would be looking a the mean impact which has the same pattern as the accumulated impact but a much smaller amplitude (see Fig. 2 for whole period compared Figs 5 and 6 in the manuscripts covering the Polarstern period). Thus, using the mean impact, instead of accumulated impact, would make the detection of significant differences between observations and simulations more difficult.

  This has been clarified in the revised manuscript (section 3) by emphasizing the importance of evaluating the accumulated impact of the mass fixer during the last month of the simulations, as opposed to the mean impact throughout the whole period.

- *The main aim of the paper is improving the inter-hemispheric gradient of total column atmospheric CO2 and CH4, however it is useful to study vertical cross sections and/or profiles of mass conservation error.*

  In the proportional mass fixer experiments, the mass fixer correction is applied uniformly throughout the column, whereas in the taylored Bermejo and Conde mass fixer, most of the correction is performed at the lower levels where the atmospheric mass and the tracer mass are largest (Fig. 3). This is already mentioned in section 2.2 of the manuscript. Further work should be done to compare the impact of the mass fixer on the vertical gradients of $CO_2$ and $CH_4$ using observations, e.g. from aircrafts, but this is out of the scope of this study.

**Technical comments**

- *Page 3, line 9: performace -> performance* Done.

- *Figures 3-6: no labels for TCCON FTS sites; TCCO2, TCCH4 should be replaced with XCO2 and XCH4 respectively.* Done. We have not been able to include the TCCON labels on the

[Figure]

Figure 1: Time series of instantaneous and accumulated global mean mass conservation error for $CO_2$ and $CH_4$ from 1 March 2013 to 30 April 2014. Low/high resolution experiments are depicted by red/blue lines.

map because they would obstruct the small plot. We have included a reference to the TCCON table listing all the sites and their lat/lon coordinates in the Figure captions instead.

- *Figures 7-10: CO2 and CH4 should be replaced with XCO2 and XCH4 respectively.* Done.

[Figure]

Figure 2: Difference in mean $XCO_2$ [ppm] and $XCH_4$ [ppb] for the period from 1 March 2013 to 30 April 2014 between (a,b) the simulations using the proportional mass fixer and the simulation without mass fixer at high resolution; and (c,d) the simulations with Bermejo and Conde (B&C) and the simulation without mass fixer at high resolution.

[Figure]

Figure 3: Vertical section of zonal mean differences in $CO_2$ and $CH_4$ from 7 March 2014 to 10 April 2014 between (a,b) the simulation with 
[revised manuscript text omitted]